# Effects of Compression of the Ulnar Artery on the Radial Artery Catheterization

**DOI:** 10.3390/jcm11185476

**Published:** 2022-09-18

**Authors:** Cho-Long Kim, Seung-Wan Hong, Seong-Hyop Kim

**Affiliations:** 1Department of Anesthesiology and Pain Medicine, Hanyang University Medical Center, Hanyang University, Seoul 04763, Korea; 2Department of Anesthesiology and Pain Medicine, Konkuk University Medical Center, Konkuk University School of Medicine, 120-1 Neungdong-ro (Hwayang-dong), Gwangjin-gu, Seoul 05030, Korea; 3Department of Infection and Immunology, Konkuk University School of Medicine, Seoul 05029, Korea; 4Department of Medicine, Institute of Biomedical Science and Technology, Konkuk University School of Medicine, Seoul 05029, Korea

**Keywords:** ulnar artery, radial artery, ultrasound-guided, catheterization

## Abstract

Background: The study was designed to evaluate the effects of compression of the ulnar artery on blood flow (BF) and internal cross-sectional area (CSAi) of the radial artery. We also evaluated the success rate and time of successful ultrasound-guided radial artery catheterization at the first attempt with or without compression of the ulnar artery. Methods: Patients were randomly allocated to the Compression group or Standard group to be treated with or without the application of ulnar artery compression, respectively. Hemodynamic stability was confirmed, and ultrasound-guided radial artery catheterization was performed. In the Compression group, an assistant compressed the ulnar artery at 5 cm above the wrist crease and the catheterization was performed after the loss of the distal ulnar artery BF. In the Standard group, the catheterization was performed without compression of the ulnar artery. Before and after the catheterization, BF and CSAi of the radial artery were evaluated. Success rate and time to successful catheterization at the first attempt were recorded. Results: BF and CSAi of the radial artery were similar in the two groups (37.5 [19.3–66] vs. 37.0 [20.6–53.7] mL/min, respectively, *p* = 0.63; 4.0 [4.0–6.0] vs. 4.0 [3.0–5.0] mm^2^, respectively, *p* = 0.095). In the Compression group, BF and CSAi were changed to 80.9 [35.9–128.5] mL/min (*p* < 0.001) and 5.0 [4.0–7.0] mm^2^ (*p* < 0.001), respectively, after compression of the ulnar artery. There was a trend that the success rate of ultrasound-guided radial artery catheterization at the first attempt was higher in the Compression group than in the Standard group (58/59 vs. 53/59, respectively, *p* = 0.05), although the difference was not statistically significant. However, the time to successful ultrasound-guided radial artery catheterization at the first attempt was significantly shorter in the Compression group than in the Standard group (34 [27–41] s vs. 46 [36–60] s, *p* < 0.001). Conclusion: Compression of the ulnar artery augmented BF and CSAi of the radial artery. It resulted in a significantly shorter success time for ultrasound-guided radial artery catheterization at the first attempt.

## 1. Introduction

Radial artery catheterization is the standard technique in cases requiring continuous systemic blood pressure monitoring [1,2,3]. The internal cross-sectional area (CSAi) is one of the most important factors for the success of radial artery catheterization, and methods to increase the CSAi of the radial artery have been shown to facilitate the procedure [4,5].

The brachial artery originating from the axillary artery is the main artery supplying blood flow (BF) to the arm. It reaches the cubital fossa and divides into its terminal branches, the radial artery, and the ulnar artery, in the forearm. These two arteries anastomose in the hand by forming two arches, the superficial palmar arch and the deep palmar arch [6]. The radioulnar circuit is a closed-loop system that provides alteration in either branch by changing the resistance in the other branch [7]. Therefore, occlusion of the ulnar artery is expected to increase BF on the radial artery in the forearm [8]. The increased BF is also expected to increase the CSAi of blood vessels [9].

We hypothesized that compression of the ulnar artery may increase BF in the radial artery, resulting in an increase in its CSAi, which would be expected to facilitate catheterization. The study was designed to evaluate the effects of compression of the ulnar artery on BF and CSAi of the radial artery. We also evaluated the success rate and time of successful ultrasound-guided radial artery catheterization at the first attempt with or without compression of the ulnar artery.

## 2. Materials and Methods

### 2.1. Study Population

After obtaining approval from the Institutional Review Board of Hanyang University Medical Center, Seoul, Korea (Reference No, 2021-08-033; date of approval, 16 September 2021) and informed consent from all patients, this study was registered at the Clinical Research Information Service, Korea Centers for Disease Control and Prevention, Ministry of Health and Welfare (KCT0006680; date of registration, 19 October 2021; http://cris.nih.go.kr). Patients requiring general anesthesia under intra-arterial systemic pressure monitoring were enrolled in this study. The exclusion criteria were as follows: (1) urgent or emergent case, (2) positive Allen test or modified Allen test, (3) recent history of puncture of the radial artery, (4) arrhythmia, and (5) any problem on the arm or wrist. Prior to induction of anesthesia, the patients were randomly allocated to the Compression group or Standard group to be treated with or without application of ulnar artery compression, respectively, by opening a sequentially numbered sealed envelope containing the randomization assignment. The allocation sequence was generated by random-permuted block randomization. All medical staff involved in patient care were blinded to the study groups. All data were collected by a trained observer who did not participate in patient care and was blinded to the study groups.

### 2.2. Anesthesia Induction

Without any premedication, anesthesia was induced after establishing routine noninvasive patient monitoring, including pulse oximetry, electrocardiography, systemic noninvasive blood pressure (NIBP) monitoring on the nondominant arm, and bispectral index (BIS). Intravenous (i.v.) administration of lidocaine (0.5 mg/kg) was followed by i.v. administration of propofol (1.5 mg/kg). Adequate mask ventilation with loss of consciousness was confirmed and rocuronium (0.6 mg/kg) was administered i.v. under peripheral neuromuscular transmission monitoring. Remifentanil was administered i.v. and sevoflurane was administered by inhalation to maintain BIS of 40–50. Tracheal intubation was performed at a train-of-four count of 0. Patients were ventilated with 40% oxygen in air. Tidal volume of 6 mL/kg, based on ideal body weight, without positive end-expiratory pressure, was utilized. Respiration was adjusted to maintain end-tidal carbon dioxide (EtCO_2_) between 35 and 40 mmHg.

### 2.3. Ultrasound-Guided Radial Artery Catheterization

After induction of anesthesia, ultrasound-guided radial artery catheterization was performed by one investigator. The investigator was an anesthesiologist with experience in hundreds of cases of ultrasound-guided radial artery catheterization. Hemodynamic stability, defined as the maintenance of NIBP within 20% of the values on arrival at operation room and mean blood pressure above 60 mmHg for 5 min, was confirmed before radial artery catheterization. Briefly, the patient was placed in the supine position with the nondominant arm abducted to 90° and supported on an arm board. The intact collateral ulnar artery BF was confirmed using modified Allen test [10,11]. The wrist joint was extended to 45° on a wrist board and fixed with tape to the arm board. The target puncture site, located where the most prominent radial artery pulsation was felt around the styloid process of the radius, was marked with a surgical marking pen. To maintain sterile conditions, the skin on the distal 1/3 of the nondominant forearm, including the wrist, was prepared with 10% chlorhexidine according to our institutional protocol. An 11 MHz linear array transducer ultrasound probe (Konica Minolta, Inc., Tokyo, Japan) was covered with a sterile sheath to prevent infection. Before catheterization, noninvasive patient monitoring, including NIBP on the nondominant arm, heart rate (HR), and BIS, were established. During the procedure, ultrasound images for BF and CSAi of the radial artery were optimally obtained by the investigator and a physician who did not involve in patient care and were blinded to the study groups. All ultrasound findings were analyzed by the physician just before the insertion of the catheter. Baseline BF and CSAi of the radial artery at the target puncture site were recorded (BF_baseline_ and CSAi_baseline_, respectively). BF was automatically calculated from CSAi and blood velocity. In Compression group, an assistant compressed the ulnar artery 5 cm above the wrist crease to block BF completely. After confirmation of loss of the distal ulnar artery BF, BF, and CSAi of the radial artery were measured at catheterization using ultrasound (BF_catheterization_ and CSAi_catheterization_, respectively). Then, ultrasound-guided radial artery catheterization using the long-axis view of the radial artery (in-plane) with a 20-gauge catheter was performed with maintenance of compression of the ulnar artery. The catheter was inserted into the radial artery at 45° under ultrasound guidance and advanced on confirmation of arterial blood regurgitation. Compression of the ulnar artery was released, and the inserted catheter was connected to a pressure transducer set. In Standard group, ultrasound-guided radial artery catheterization was performed without compression of the ulnar artery.

The time to successful ultrasound-guided radial artery catheterization at the first attempt was defined as the time from skin insertion of the 20-gauge catheter to confirmation of arterial waveform on the monitor. Failure was defined as lack of confirmation of arterial blood regurgitation or the arterial waveform. Success times over 120 s were also defined as failures. Success rate and time to successful ultrasound-guided radial artery catheterization at the first attempt were recorded. To prevent any bias, the procedure was performed after the completion of all preparations, including positioning for the catheterization, sterile skin preparation, preparation of all items for the catheterization, and sterile ultrasound-related preparation for optimal images.

Assistant was a nurse for anesthesia and assisted all processes for ultrasound-guided radial artery catheterization. The assistant helped the compression of the ulnar artery and the connection of the radial artery catheter to the monitoring device in Compression group. The assistant helped the connection of the radial artery catheter to the monitoring device without the compression of the ulnar artery in Standard group. To facilitate compression of the ulnar artery without any hindrance during the procedure, the assistant stood at the opposite site for the catheterization. For example, when the right radial artery catheterization was performed in Compression group, the assistant stood close to the left side of the operation table, bent her back, and compressed the right ulnar artery.

### 2.4. Statistical Analysis

The primary outcomes were BF and CSAi of the radial artery. In a pilot study in 14 patients, BF and CSAi of the radial artery changed from 32.5 ± 20.2 to 45.3 ± 25.8 mL/min and from 3.4 ± 1.0 mm^2^ to 4.1 ± 1.2 mm^2^, respectively, after loss of the distal ulnar artery BF. Based on a power of 0.9 and alpha of 0.05, the sample sizes for each group were calculated as 57 for BF and 59 for CSAi of the radial artery. The secondary outcome was time to successful ultrasound-guided radial artery catheterization at the first attempt. In a pilot study in 14 patients, the times to successful ultrasound-guided radial artery catheterization were 46 ± 16 s and 55 ± 16 s for Compression and Standard groups (seven patients each), respectively. Based on a power of 0.9 and alpha of 0.05, the required sample size for each group was calculated as 55.

Statistical analyses were performed using SPSS for Windows (ver. 27.0; SPSS Inc., Chicago, IL, USA). Categorical variables were analyzed using the chi-square or Fisher’s exact test, and continuous variables were compared between two groups using the independent *t*-test or Mann–Whitney U test. Changes in continuous variables after ulnar artery compression within Compression group were analyzed using the paired *t*-test or Wilcoxon matched paired *t*-test. Times to successful ultrasound-guided radial artery catheterization at the first attempt were compared by the Kaplan–Meier method. Categorical variables were expressed as number of patients. Continuous variables were expressed as the mean ± standard deviation or median (interquartile range). In all analyses, *p* < 0.05 was taken to indicate statistical significance.

## 3. Results

A total of 118 patients treated between October 2021 and April 2022 were enrolled in the study and were evenly randomized into the two groups without any exclusions or loss to follow-up. The demographic data, including amounts of anesthetic agents from anesthesia induction to ultrasound-guided radial artery catheterization, were similar in the two groups (Table 1). The hemodynamic and respiratory parameters at ultrasound-guided radial artery catheterization also had similar values in the two groups (Table 2).

BF_baseline_ and CSAi_baseline_ of the radial artery were similar in the Compression and Standard groups (37.5 [19.3–66] vs. 37.0 [20.6–53.7] mL/min, respectively, *p* = 0.63; 4.0 [4.0–6.0] vs. 4.0 [3.0–5.0] mm^2^, respectively, *p* = 0.095). In the Compression group, BF and CSAi were changed to 80.9 [35.9–128.5] mL/min (*p* < 0.001) and 5.0 [4.0–7.0] mm^2^ (*p* < 0.001), respectively, after compression of the ulnar artery (Figure 1).

There was a trend that the success rate of ultrasound-guided radial artery catheterization at the first attempt was higher in the Compression group than in the Standard group (58/59 vs. 53/59, respectively, *p* = 0.05), although the difference was not statistically significant (Table 3). However, the time to successful ultrasound-guided radial artery catheterization at the first attempt was significantly shorter in the Compression group than in the Standard group (34 [27–41] s vs. 46 [36–60] s, *p* < 0.001) (Table 3 and Figure 2).

## 4. Discussion

The results of the present study showed that compression of the ulnar artery increased BF and CSAi of the radial artery, leading to a shorter time to successful ultrasound-guided radial artery catheterization with a trend of higher success rate at the first attempt.

Many important factors, including proper positioning and selection of the catheterization site with the strongest pulse, are associated with the success of the radial artery catheterization [12,13,14]. Several trials showed that increasing the CSAi of the radial artery is the best way to improve the success rate and time to successful radial artery catheterization [4,5,15,16,17,18,19]. Subcutaneous nitroglycerin and nitroprusside administration at the site of radial artery catheterization has been shown to be suitable for increasing the CSAi of the radial artery [5,18]. Oudhour et al. reported that local infiltration with dinitrate isosorbide and lidocaine rather than lidocaine alone in the transradial approach for coronary artery angiography reduced the number of punctures and the time to successful radial artery catheterization without any change in the severity of pain associated with the procedure [20]. Moreover, Majure et al. reported that lidocaine did not alter the effect of nitroglycerin on the increase in CSAi of the radial artery [17]. However, trials using drugs have risks of adverse events, such as needle-related problems and specific drug-related effects. Therefore, we wished to develop a practical and simple method that would facilitate radial artery catheterization.

As ultrasound-guided radial artery catheterization has been recently popular, the confirmation of intact collateral circulation, using ultrasound, has been standardized. However, before the advent of ultrasound, the Allen test or modified Allen test was the first step to performing radial artery catheterization for the determination of the sufficiency of the collateral ulnar artery BF to the hand. It meant that this was not a new trial of compression of the ulnar artery, but an evaluation of a routine procedure. Zhou et al. reported that compression of the ulnar artery for 30 min increased both the radial artery diameter and the success rate of the radial artery catheterization [4]. Yilmaztepe and Yilmaz reported that compression of the ulnar artery for 1 min was sufficient to increase the diameter of the radial artery [15]. Unlike previous studies, we examined BF and CSAi of the radial artery. The increased CSAi of the radial artery was due to the increased BF of the radial artery caused by occlusion of the collateral artery during ulnar artery compression. The increased BF with increased CSAi of the radial artery may have occurred immediately after the occlusion of the collateral artery. Therefore, the duration of ulnar artery compression was not important to increase BF and CSAi of the radial artery.

In the present study, the radial artery in the nondominant hand was chosen to minimize patient discomfort postoperatively. However, the degree of radial artery BF augmentation would have been different if the study was conducted with the dominant hand, not with the nondominant hand as in the present study because the ulnar artery is anatomically larger in the dominant hand than in the nondominant hand [9].

The present study was conducted by a single investigator. Considering the characteristics of the study, the proficiency of the investigator could have increased over time. Moreover, although the investigator was blinded to the study groups, the possibility of subtle bias due to subjective factors remained. However, the objective measurement of BF and CSAi of the radial artery may have minimized the effects of such subjective factors. Therefore, the Kaplan–Meier method was used to examine the time to successful ultrasound-guided radial artery catheterization at the first attempt in the present study.

As mentioned above, augmenting BF of the radial artery with increased CSAi of the radial artery was important to obtain the success of ultrasound-guided radial artery catheterization. The present study demonstrated that compression of the ulnar artery at ultrasound-guided radial artery catheterization augmented BF of the radial artery with increased CSAi of the radial artery. It finally resulted in a significantly shorter success time and a trend of the increased success rate of ultrasound-guided radial artery catheterization at the first attempt. In previous studies, new techniques for augmenting BF or increasing CSAi of the radial artery have shown superior to conventional procedures, in the aspect of success time or success rate [4,5,15]. It meant that ultrasound-guided radial artery catheterization, including the compression of the ulnar artery, leading to augmented BF or increasing CSAi of the radial artery, was superior to ultrasound-guided radial artery catheterization without compression of the radial artery. Moreover, the new technique, utilization of just routine process of the procedure, in the present study did not need any specific tool or agent as in previous studies.

In conclusion, compression of the ulnar artery augmented BF and CSAi of the radial artery. It resulted in a significantly shorter success time for ultrasound-guided radial artery catheterization at the first attempt.

## Figures and Tables

**Figure 1 jcm-11-05476-f001:**
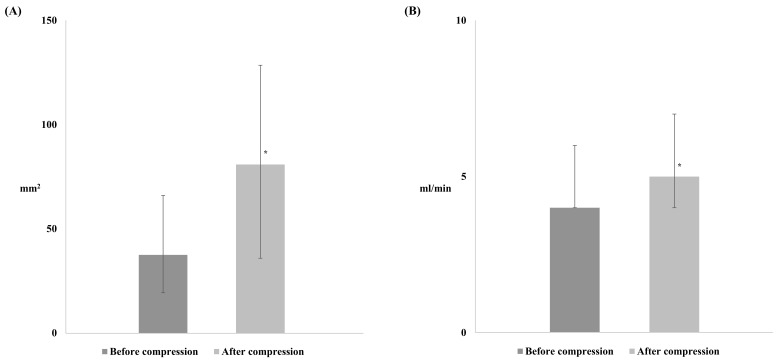
The changes in the blood flow (BF) and the internal cross-sectional area of the radial artery before and after the compression of the ulnar artery in Compression group. (**A**) BF, (**B**) CSAi. * *p* < 0.05 compared with before compression.

**Figure 2 jcm-11-05476-f002:**
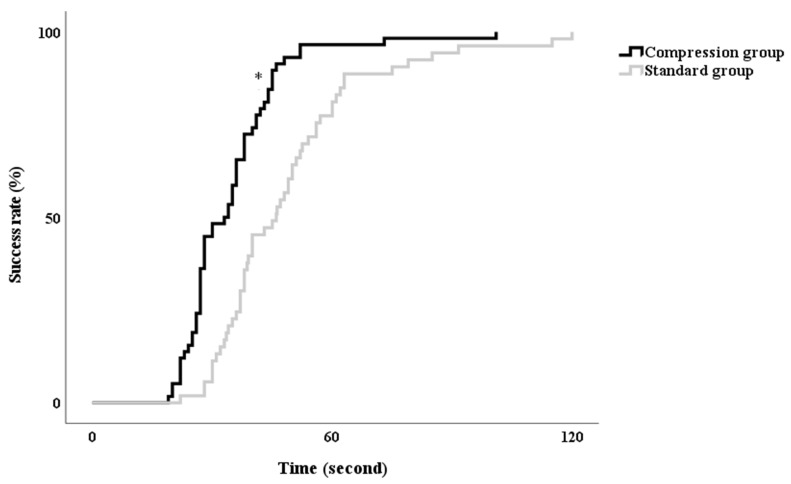
Kaplan–Meier estimator for success time of the radial artery catheterization. * *p* < 0.05 compared with Standard group.

**Table 1 jcm-11-05476-t001:** Demographic data.

		Compression Group	Standard Group	*p* Value
Gender (Male/Female)		31/28	30/29	0.850
Age (yrs)		67 [55–74]	68 [56–76]	0.483
Height (cm)		167.0 [156.0–172.3]	165.0 [165.0–170.0]	0.888
Weight (kg)		67.0 [59.0–75.0]	64.0 [56.0–76.0]	0.519
ASA PS				0.800
	I	19	19	
	II	28	26	
	III	12	14	
Anesthetic agents				
	Propofol (mg)	90.0 [80.0–100.0]	90.0 [70.0–100.0]	0.233
	Remifentanil (μg)	50.0 [40.0–60.0]	50.0 [40.0–60.0]	0.819
	Rocuronium (mg)	50.0 [40.0–50.0]	50 [40.0–50.0]	0.881
	Sevoflurane (vol%)	1.9 [1.7–2.0]	1.9 [1.8–2.0]	0.800

Data are expressed as number of patients, mean **±** standard deviation or median (interquartile range). Abbreviations: ASA PS, American Society of Anesthesiologists Physical Status.

**Table 2 jcm-11-05476-t002:** Hemodynamic and respiratory parameters at ultrasound-guided radial artery catheterization.

		Compression Group	Standard Group	*p* Value
Hemodynamic parameters				
	BIS	49 [43–56]	49 [45–55]	0.783
	Systolic BP (mmHg)	127 ± 20	125 ± 26	0.808
	Diastolic BP (mmHg)	86 ± 14	84 ± 15	0.576
	Mean BP (mmHg)	96 [83–105]	92 [81–103]	0.292
	HR (beat/min)	84 [71–96]	82 [69–95]	0.363
Respiratory parameters				
	TV (mL/kg)	402 [354–450]	384 [336–456]	0.519
	RR (frequency/m)	11 [10–12]	12 [10–12]	0.754
	PIP (cmH_2_O)	12 [10–13]	12 [10–13]	0.775
	Pplat (cmH_2_O)	11 [10–12]	11 [9–13]	0.798
	P_Et_CO_2_ (mmHg)	38 [36–39]	37 [36–39]	0.977

Data are expressed as number of mean ± standard deviation or median (interquartile range). Abbreviations: BIS, bispectral index; Sys, systemic non-invasive systolic; Dia, systemic non-invasive diastolic; BP, blood pressure; HR, heart rate; TV, tidal volume; RR, respiratory rate; PIP, peak inspiratory pressure; Pplat, plateau inspiratory pressure; P_Et_CO_2_, partial pressure of end-tidal carbon dioxide.

**Table 3 jcm-11-05476-t003:** Success rate and time of ultrasound-guided radial artery catheterization at the first attempt.

Variables	Compression Group	Standard Group	*p* Value
Success rate	58/59	53/59	0.05
Success time (second)	34 [27–41]	46 [36–60]	<0.001

Data are expressed as mean ± standard deviation or median (interquartile range).

## Data Availability

The data presented in this study are available on request from the corresponding author.

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
