# Peer review of "Effects of Compression of the Ulnar Artery on the Radial Artery Catheterization"

_jcm, 2022, doi:10.3390/jcm11185476_

Round 1
Reviewer 1 Report
This is a nice little paper describing a potential improvement to a technical procedure common in anesthesia. I have a few comments to offer for consideration:
A few comments came to mind as I reviewed it.
Methods: -I think that it is not always felt that arterial pressure monitoring is particularly "invasive" in common use of that term. As such the phrase in the sentence "Patients requiring general anesthesia under invasive systemic artery pressure monitoring..." is awkward in my view: how about 'patients requiring ga with intra-arterial pressure monitoring' as a simple and adequate description.
-the phrase 'Allen test or modified Allen test' is used several times. Did the authors really do the Allen test as described (https://www.ncbi.nlm.nih.gov/books/NBK507816/)? I bet not - I've actually never seen it done tht way. In any case for simplicity I think it would be quite adequate to use the simple phrase 'modified Allen test'. In actual fact I'm not really sure quite why the authors used the Allen test - surely as an ultrasound-based study one might just have well simply used ultrasound to determine palmar arch continuity.... The value of the Allen test has been questioned as noted in the quoted reference. However I won't quibble too much - if the authors wanted to do the Allen test, whether or not the reader might question it's usefulness, then that's OK.
-the authors could comment in a bit more detail about the logistics of the assistant compressing the ulnar artery. Wasn't that assistant in the way of the operator during the line insertion - and if not how that was arranged could be helpful.
-the authors refer to blinding of "all medical staff" and "trained observer". Does this mean the operator (arterial line inserter) was blinded too? Was the assistant providing compression placed during all procedures - compressed and not compressed?
-That point may not be terribly important but is interesting in that the time measurement outcome variable was stated to be "defined as the time from skin insertion of the 20-gauge catheter to confirmation of arterial waveform on the monitor". If there was an assistant in the way, did working around that assistant, getting set up with sterile setup/ultrasound probe positioning/operator positioning vary between the two groups of procedures? It is stated that the times were different between the two techniques but if the "preparation time" was longer for the more complex procedure then this would be an important point.
-the authors do not state how the ultrasound images were analyzed to produce the figures with respect to blood velocity and cross-sectional area. Were they stored and analyzed "off-line" by a blinded expert/experience analyst?
Results:
-the authors state, "The success rate of ultrasound-guided radial artery catheterization at the first attempt was higher in Compression group than in Standard group (58/59 vs. 53/59, respectively, p = 0.05), although the difference was not statistically significant". If the quoted difference "was not statistically significant" then there was no difference (you can't choose to use statistical analysis in part of your sentence and not in a different part. At best the authors could say 'there was a trend' but this is considered a weak comment...
-in any case the time of insertion as defined was indeed significant (statistically and otherwise).
Discussion:
-again, the authors stage that 'compression... leads to a higher success rate' which they cannot state from their results.
-they state that 'Allen test should be performed to determine'... which as I have noted is at best a controversial statement.
-the authors state the nondominant had was chosen to minimize patient discomfort. I'm not clear about this as the patients were under general anesthesia, and postoperatively whether there would be a difference in discomfort between the non-dominant and dominant hand.
-the final conclusion, that compression of the ulnar artery may be recommended as a standard step, is somewhat strong. "Useful step" might be a better term. Establishing a technique as a "standard" is one which one should do only very carefully.
Author Response
At first, I thank the editors and referees of the “Journal of Clinical Medicine” for taking their times to review of my paper, entitled “Effects of compression of the ulnar artery on the radial artery catheterization”.
I have made some corrections and clarifications in the manuscript after going over the referee’s comments. The changes are summarized below and the corrected or newly added sentences were expressed with red-color in the manuscript.
Reviewer-1
- As such the phrase in the sentence "Patients requiring general anesthesia under invasive systemic artery pressure monitoring..." is awkward in my view: how about 'patients requiring ga with intra-arterial pressure monitoring' as a simple and adequate description.
: As reviewer’s recommendation, we changed “invasive systemic artery pressure monitoring” into “intra-arterial systemic pressure monitoring”. We deleted “invasive” but remained “systemic” to distinguish from pulmonary arterial pressure monitoring.
- In any case for simplicity I think it would be quite adequate to use the simple phrase 'modified Allen test'.
: As reviewer’s recommendation, we changed “Allen test or modified Allen test” into “modified Allen test”
- the authors could comment in a bit more detail about the logistics of the assistant compressing the ulnar artery. Wasn't that assistant in the way of the operator during the line insertion - and if not how that was arranged could be helpful.
: To facilitate compression of the ulnar artery without any hindrance at the procedure, the assistant stood at the opposite site for the catheterization. For example, when the right radial artery catheterization was performed in Compression group, the assistant stood close to the left side of the operation table, bent her back and compressed the right ulnar artery.
- We described above in Materials and Methods.
- the authors refer to blinding of "all medical staff" and "trained observer". Does this mean the operator (arterial line inserter) was blinded too? Was the assistant providing compression placed during all procedures - compressed and not compressed?
: Yes, ultrasound-guided radial artery catheterization was performed by one investigator as described in Materials and Methods. The investigator was not blinded. In the study, assistant was a nurse for anesthesia. The assistant helped the compression of the ulnar artery and the connection of the radial artery catheter to the monitoring device in Compression group. The assistant helped the connection of the radial artery catheter to the monitoring device without the compression of the ulnar artery in Standard group.
We described above in Materials and Methods.
- If there was an assistant in the way, did working around that assistant, getting set up with sterile setup/ultrasound probe positioning/operator positioning vary between the two groups of procedures? It is stated that the times were different between the two techniques but if the "preparation time" was longer for the more complex procedure then this would be an important point.
: Yes, we absolutely agreed reviewer’s comment. Radial artery catheterization was performed after the completion of all preparation to prevent any bias.
We described above in Materials and Methods.
- To prevent any bias, the procedure was performed after the completion of all preparations, including positioning for the catheterization, sterile skin preparation, preparation of all items for the catheterization and sterile ultrasound-related preparation for optimal images.
- the authors do not state how the ultrasound images were analyzed to produce the figures with respect to blood velocity and cross-sectional area. Were they stored and analyzed "off-line" by a blinded expert/experience analyst?
: Obtaining optimal images for blood flow and internal cross-sectional area of the radial artery were performed by collaboration of the investigator and a physician who did not involve in patient care and was blinded to the study groups. All ultrasound findings were analyzed by the physician just before the insertion of the catheter.
We described above in Materials and Methods.
- During the procedure, ultrasound images for BF and CSAi were optimally obtained by the investigator and a physician who did not involve in patient care and was blinded to the study groups. All ultrasound findings were analyzed by the physician just before the insertion of the catheter.
- "The success rate of ultrasound-guided radial artery catheterization at the first attempt was higher in Compression group than in Standard group (58/59 vs. 53/59, respectively, p = 0.05), although the difference was not statistically significant". If the quoted difference "was not statistically significant" then there was no difference (you can't choose to use statistical analysis in part of your sentence and not in a different part. At best the authors could say 'there was a trend' but this is considered a weak comment...
: As reviewer’s recommendation, we change the sentence as below.
- There was a trend that the success rate of ultrasound-guided radial artery catheterization at the first attempt was higher in Compression group than in Standard group (58/59 vs. 53/59, respectively, p = 0.05), although the difference was not statistically significant.
- in any case the time of insertion as defined was indeed significant (statistically and otherwise).
: Yes, success time was significantly shorter in Compression group, compared with Standard group, although the first success rate did not have any significant difference. The finding, we thought, was clinically meaningful.
- again, the authors stage that 'compression... leads to a higher success rate' which they cannot state from their results.
: As the reply for query #7, we changed the sentence as below.
- The results of the present study showed that compression of the ulnar artery increased BF and CSAi of the radial artery, leading to shorter time to successful ultrasound-guided radial artery catheterization with a trend of higher success rate at the first attempt.
- they state that 'Allen test should be performed to determine'... which as I have noted is at best a controversial statement.
: Although reviewer-1 expressed skeptical viewpoint for Allen test just before radial artery catheterization, Allen test is the standard process just before the start of the radial artery catheterization. As reviewer issued the controversy for Allen test, we changed the sentence as below.
- As ultrasound-guided radial artery catheterization has been recently popular, the confirmation of intact collateral circulation, using ultrasound, has been standardized. However, before the advent of ultrasound, Allen test or Modified Allen test was the first step to perform radial artery catheterization for the determination of the sufficiency of the collateral ulnar artery BF to the hand.
- the authors state the nondominant had was chosen to minimize patient discomfort. I'm not clear about this as the patients were under general anesthesia, and postoperatively whether there would be a difference in discomfort between the non-dominant and dominant hand.
: We were sorry for the confused paragraph. Nondominant hand was chosen for the catheterization was to minimize the patient discomfort postoperatively. In the present study, we did not compare augmentation degree of radial artery BF at compression of ulnar artery between dominant and nondominant hands. However, the size of the ulnar artery is anatomically larger in dominant hand than nondominant hand. Therefore, compression of the ulnar artery in dominant hand than nondominant was expected to increase more blood flow and internal cross-sectional area of the radial artery. We changed the sentences to understand the meaning easily.
- In the present study, the radial artery in the nondominant hand was chosen to minimize patient discomfort postoperatively. However, the degree of radial artery BF augmentation would have been different if the study was conducted with the dominant hand, not with the nondominant hand as the present study, because the ulnar artery is anatomically larger in the dominant hand than the nondominant hand.
- the final conclusion, that compression of the ulnar artery may be recommended as a standard step, is somewhat strong. "Useful step" might be a better term. Establishing a technique as a "standard" is one which one should do only very carefully.
: As reviewer-1’s recommendation, we changed the sentence as below.
- Therefore, compression of the ulnar artery may be recommended as an useful step for ultrasound-guided radial artery catheterization.
I hope the revised manuscript will better meet the requirements of the “Journal of Clinical Medicine” for publication.
I thank you again for the constructive review by the referees.
Sincerely yours,
Seong-Hyop Kim, M.D., Ph.D.
Reviewer 2 Report
The authors present a simple and instructive study with the potential to improve a successful rate of a routinely used procedure - radial artery cannulation.
Overall the study is of good quality.
I have no major comments.
Minor comments:
1) paragraph 2.2. – why was lidocaine used? To blind the circulatory reflex to intubation? This is just for me, you do not need to justify this in the text as I am curious.
2) dtto – why did you use zero PEEP? This is not recommended.
3) dtto – ETCO... -> EtCO... (small t)
4) paragraph 2.3. – the hemodynamic stability should be defined
5) dtto – describe the Modified Allen test or give a proper citation
6) dtto – was the single investigator an experienced clinician? This should be stated.
7) paragraph 2.4. – I do not understand why do you give the results of a pilot study of your study in statistics. Any kind of results should be in the results section. This is quite confusing. Please take care of this adequately.
8) dtto – how was the normality of data tested?
9) paragraph 4 – please add information about the effect of sole lidocaine on BF/CSAi and if these data are published give a citation.
Author Response
At first, I thank the editors and referees of the “Journal of Clinical Medicine” for taking their times to review of my paper, entitled “Effects of compression of the ulnar artery on the radial artery catheterization”.
I have made some corrections and clarifications in the manuscript after going over the referee’s comments. The changes are summarized below and the corrected or newly added sentences were expressed with red-color in the manuscript.
Reviewer-2
- why was lidocaine used? To blind the circulatory reflex to intubation? This is just for me, you do not need to justify this in the text as I am curious.
: As described in Materials and Methods, propofol was used for anesthesia induction. The purpose for the use of lidocaine at anesthesia induction was mainly the prevention of pain at propofol administration.
- why did you use zero PEEP? This is not recommended.
: We performed protective ventilation in all patients. We confirmed inspiratory pressure just after mechanical ventilation and applied positive end expiratory pressure (PEEP). It meant that protective ventilation without PEEP was temporarily applied at anesthesia induction.
- .. -> EtCO
: We changed it.
- the hemodynamic stability should be defined
: Ultrasound-guided radial artery catheterization was performed after anesthesia induction. As reviewer-2 knew, hemodynamic fluctuation was frequently occurred during anesthesia induction and had the chance to influence the success of ultrasound-guided radial artery catheterization. Therefore, ultrasound-guided radial artery catheterization was performed after confirmation of hemodynamic stability. Hemodynamic stability was defined as the maintenance of non-invasive blood pressure (NIBP) within 30% of the values on arrival at operation room for 5 minutes.
We described as below in Material and Methods.
- Hemodynamic stability, defined as the maintenance of NIBP within 20% of the values on arrival at operation room and above mean blood pressure above 60 mmHg for 5 minutes, was confirmed before radial artery catheterization.
- describe the Modified Allen test or give a proper citation
: We added the references.
- was the single investigator an experienced clinician? This should be stated.
: The investigator was an anesthesiologist with experience of hundreds cases of ultrasound-guided radial artery catheterization.
We explained the investigator in Materials and Methods.
- I do not understand why do you give the results of a pilot study of your study in statistics. Any kind of results should be in the results section. This is quite confusing. Please take care of this adequately.
: Pilot study was conducted just to get the sample size. As institutional guideline, the pilot study was conducted after obtaining separate IRB approval from the present study. The results from the pilot study were not included in the present study. Therefore, we described the results from the pilot study in Statistics.
- how was the normality of data tested?
: We checked normal distribution of data. The results with normal distribution were expressed as mean ± standard deviation and the results without normal distribution were expressed as median [interquartile range].
We changed above in Materials and Methods, Results, Tables and Figure 1.
- please add information about the effect of sole lidocaine on BF/CSAi and if these data are published give a citation.
: At first, we did not use local infiltration of lidocaine at ultrasound-guide radial artery catheterization in the present study and did not have data for local infiltration of lidocaine. It was difficult to find any reference about the effect of local infiltration with lidocaine on blood flow (BF) or internal cross-sectional area (CSAi) of the artery for catheterization, although there were several reports that nerve block with lidocaine increased BF and CSAi of the artery.
I hope the revised manuscript will better meet the requirements of the “Journal of Clinical Medicine” for publication.
I thank you again for the constructive review by the referees.
Sincerely yours,
Seong-Hyop Kim, M.D., Ph.D.